# Rotation Active Sensors Based on Ultrafast Fibre Lasers

**DOI:** 10.3390/s21103530

**Published:** 2021-05-19

**Authors:** Igor Kudelin, Srikanth Sugavanam, Maria Chernysheva

**Affiliations:** 1Aston Institute of Photonic Technologies, Aston University, Birmingham B4 7ET, UK; 2School of Computing and Electrical Engineering, IIT Mandi, Kamand, Himachal Pradesh 175075, India; ssrikanth@iitmandi.ac.in; 3Leibniz Institute of Photonic Technology, Albert-Einstein str. 9, 07745 Jena, Germany; maria.chernysheva@leibniz-ipht.de

**Keywords:** mode-locked laser, fibre gyroscope, Dispersive Fourier Transform, fibre sensors

## Abstract

Gyroscopes merit an undeniable role in inertial navigation systems, geodesy and seismology. By employing the optical Sagnac effect, ring laser gyroscopes provide exceptionally accurate measurements of even ultraslow angular velocity with a resolution up to 10${}^{-11}$ rad/s. With the recent advancement of ultrafast fibre lasers and, particularly, enabling effective bidirectional generation, their applications have been expanded to the areas of dual-comb spectroscopy and gyroscopy. Exceptional compactness, maintenance-free operation and rather low cost make ultrafast fibre lasers attractive for sensing applications. Remarkably, laser gyroscope operation in the ultrashort pulse generation regime presents a promising approach for eliminating sensing limitations caused by the synchronisation of counter-propagating channels, the most critical of which is frequency lock-in. In this work, we overview the fundamentals of gyroscopic sensing and ultrafast fibre lasers to bridge the gap between tools development and their real-world applications. This article provides a historical outline, highlights the most recent advancements and discusses perspectives for the expanding field of ultrafast fibre laser gyroscopes. We acknowledge the bottlenecks and deficiencies of the presented ultrafast laser gyroscope concepts due to intrinsic physical effects or currently available measurement methodology. Finally, the current work outlines solutions for further ultrafast laser technology development to translate to future commercial gyroscopes.

## 1. Introduction

The term “gyroscope” was introduced by Léon Foucault in 1852, referring to a device for measuring angular rotation. The gyroscope, built by Léon Foucault, helped estimate the rotation velocity of Earth, which showed the potential for the use of the technology. At that time, gyroscopes were entirely mechanical and based on rotating a massive sphere [1] or disk [2]. Later, gyroscopes started utilising electric motors to maintain the rotation of the moving parts of the device. Nowadays, principal types of gyroscopes include mechanical, micro-electro-mechanical system (MEMS), quartz and laser gyroscopes [3]. Mechanical gyroscopes provide high performance with good stability [3]; however, the presence of a moving part, relatively low start-up time and high cost limit their applications.

In 1963, two years after the laser invention, the first laser gyroscope was demonstrated by Macek and Davis [4]. As the name implies, the laser gyroscope uses a ring laser cavity as a rotation sensing element. The underlying basis of light response to angular movements relies on the Sagnac effect or Sagnac interference, originally proposed by Georges Sagnac in 1913 [5,6]. According to it, optical waves, counter-propagating inside the cavity, experience the difference in optical paths due to the rotation of the gyroscopic platform. Therefore, laser gyroscopes assess the angular velocity by evaluating the time delay between counter-propagated beams, which is also manifested as a shift of the pulse phase or the change of oscillation frequencies (Section 2). Nowadays, modern laser gyroscopes provide the most accurate rotation measurements and can detect precise rotational motions associated with seismic events [7], Chandler and Annual wobbles [8].

A related device, also operating based on the Sagnac effect, is a fibre optic gyroscope (FOG), as shown in Figure 1a. The first FOG was demonstrated in 1973 [9]. FOG presents a passive interferometer (in other words, one that does not contain active media), where two laser light fields counter-propagate inside an optical fibre loop. FOG detects a phase shift between the laser beams to define the Sagnac effect. High mechanical flexibility and low losses allow coiling of long optical fibres for FOG’s sensitivity enhancement. In such a way, the overall accumulated optical Sagnac effect is multiplied by the number of turns [10]. Due to reasonably precise rotational rate information and rigid design [11], FOGs are currently widely applied in inertial navigation systems [12,13].

The most severe limitation of active laser gyroscopes is a ’lock-in’ effect. This effect originates from the interaction of the counter-propagating beams inside a laser cavity due to back-scattering at reflective surfaces [14] and amplification in active media. When the laser gyroscope rotates, such interaction can cause frequencies synchronisation of counter-propagating beams. The range of angular velocities below critical value $\Omega_{L}$ cannot be detected and is called a dead-band. Therefore, the lock-in effect is particularly severe at low rotation velocities. The dead-band can be suppressed, but not completely eliminated, by coating cavity mirrors with high-reflective dielectrics [15], by applying external dithering [16] or improvement of laser cavity geometry (e.g., developing triangle-shaped cavity) [17]. However, such methods substantially increase the complexity and total price of the laser gyroscopes. In these regards, the compact and maintenance-free all-fibre configuration becomes less advantageous for application in laser gyroscopes due to inherent Rayleigh scattering [18].

An alternative approach suggested for the lock-in effect mitigation is the exploitation of ultrashort pulses instead of continuous-wave radiation in optical gyroscopes. Ultrashort pulses feature a small spatial length (∼0.3μm per fs) and, thus, interact shortly only in two points in the cavity or fibre loop; therefore, they are less subjected to lock-in synchronisation [19]. In ultrafast laser gyroscopes, the low-intensity back-scattered light is further suppressed in a saturable absorber used for pulse formation (more details in Section 3.1). The first ultrafast laser gyroscope was realised based on bidirectional ring dye laser in 1991 by Dennis et al. [20]. Later in the same year, two groups demonstrated independent studies of the Sagnac effect in bidirectional mode-locked dye laser [21] and bidirectional picosecond diode ring laser [22]. The latest works on gyroscopic measurements in bidirectional mode-locked laser showed a sensitivity of 10${}^{-8}$ rad [23,24]. These works confirm a significant reduction of the dead-band zone, which induced significant research and development interest in ultrafast lasers as a platform for gyroscopic measurements.

In 1993, Jeon et al. realised the first fibre-based mode-locked laser gyroscope with an intracavity Sagnac interferometer [25]. Ultrafast fibre lasers have been intensively studied over the past few decades and proved to be an excellent tool for accurate measurements in absolute length and frequency metrology [26,27] and spectroscopy [28]. However, today only a few works have been presented related to gyroscopic measurements via mode-locked fibre lasers. Nevertheless, the latest work establishes a novel approach for interferometric measurements of the Sagnac effect based on time-stretch technique of ultrashort pulses, which allows the achievement of the highest acquisition rates in the MHz range [29]. Generally, ultrashort pulsed fibre laser gyroscopes inherit exceptional perspectives to become a prominent tool for dead-band-free measurements of angular velocities.

Previous review papers are focused on laser gyroscopes and their applications [3,17,30]. Given recent trends, this Review is dedicated to the development of ultrashort pulsed fibre laser gyroscopes over the last years to foresee future technological and scientific advancement of the field. Portraying the whole process from fabrication and characterisation of bidirectional mode-locked fibre laser cavities to the optical Sagnac effect evaluation, we provide insight into the possibilities of such gyroscopic systems, hence, deliver a complete interdisciplinary package for future research or an introduction to the field. The outline of the Review is the following. The first section is focused on the basics of the Sagnac effect, architectures of laser gyroscopes and their limitations with up-to-date performance. In the second section, we cover the fundamentals of mode-locking operation, characterisation of pulsed laser generation, including bidirectional fibre laser setups for potential gyroscopic applications. The last section encompasses the up-to-date works of gyroscopic measurements in ultrafast fibre lasers. The Review is concluded with an overview of the current research directions and outcomes with future perspectives.

## 2. Sagnac Effect

As mentioned above, optical gyroscopes are based on the Sagnac effect that relates the difference in optical paths of counter-propagating beams ΔL to the angular velocity of the gyro platform. If the laser gyroscope platform rotates at an angular velocity $\vec{\omega }$ rad/s, the difference in optical paths between counter-propagating beams could be expressed:(1)$$\Delta L=\frac{4\vec{A}}{c}\vec{\omega }\;$$
where $\vec{A}$ is an oriented area of the laser gyroscope and *c* is the speed of light in media with refractive index *n* (n≈1.47 for conventional silica-based fibres). The vector form indicates that the laser gyroscopes are sensitive only to the rotation in the laser platform’s plane. The difference in optical paths leads to a different time for counter-propagating beams to make a round trip through the laser gyroscope cavity:(2)$$\Delta t=\frac{4\vec{A}}{c^{2}}\vec{\omega }\;$$

The Sagnac effect can also be expressed as a Sagnac-induced phase shift ($\phi =\frac{2\pi c}{\lambda_{c}}\;\cdot \;t$) between counter-propagating beams:(3)$$\Delta \phi =\frac{8\pi \vec{A}}{\lambda_{c}c}\vec{\omega }\;$$
where $\lambda_{c}$ is the carrier wavelength of the beams. The presented Sagnac Equations (Equation 1)–(Equation 3) give a linear relationship between rotation velocity and parameters of the laser beams and could be applied for arbitrary gyroscope geometry. The fraction which defines the ratio between angular velocity and the beam parameter is known as a scale factor. The Sagnac effect is perfectly explained in the context of general relativity. The complete derivation of the Sagnac formula with special-relativistic analysis and corrections can be found in [31,32].

A laser cavity maintains generation only at integer numbers of propagating wavelengths L=n·λ, which present longitudinal modes. Thus, according to the previous statement, the difference in the total optical paths ΔL caused by the Sagnac effect can be manifested through the change of oscillation frequencies Δν or beam wavelengths. The associated beat-note frequency or ‘*Sagnac frequency*’ can be derived as:(4)$$\frac{\Delta \nu }{\nu }=\frac{\Delta L}{L}\;\Delta \nu =\frac{\Delta L\;\cdot \;\nu }{L}=\frac{4\vec{A}}{L\lambda_{c}}\vec{\omega }\;$$
where ν is an initial oscillation frequency and *L* is the cavity perimeter. The equation illustrates that longitudinal frequency modes could be slightly shifted with the rotation of the platform. The associated frequency shift can also be interpreted in terms of a Doppler shift, caused by moving the laser platform [23].

The introduced equations for the Sagnac effect establish a basis for angular velocity measurements in laser gyroscopes. However, there are several modifications to increase the response of the laser cavity to the Sagnac effect. For instance, in 1981, A. Kaplan and P. Meystre proposed the implementation of a nonlinearly induced non-reciprocity [33]. Other works applied the concept of a ’slow light’ [34] by introducing a giant dispersion to increase the Sagnac effect [24,35,36,37,38,39].

### 2.1. Methods for Sagnac Effect Detection

As discussed briefly above, optical interferometers for Sagnac effect measurements are divided into passive and active. In passive interferometers, an external laser source is needed (Figure 1a). In contrast, active sensors comprise a laser cavity, i.e., interferometer with an active media (Figure 1b). The laser gyroscope has to maintain a generation of two counter-propagating beams, which are superimposed at the output to retrieve Sagnac-induced relative changes.

Generally, the Sagnac effect can be detected in the temporal domain as a difference between arriving time of counter-propagated beams (Equation (Equation 2)) , in the frequency domain as a beat-note (Equation (Equation 3)) or as a Sagnac-induced phase shift between counter-propagating beams. However, even the fastest photodetectors have limited rise time in the ps range, which also strictly restrain the overall resolution of the gyroscopes in the time domain.

An interferometric pattern of combined counter-propagating beams on the photodetector provides phase-sensitive measurements. During gyroscope platform rotation, counter-propagating beams experience additional Sagnac-induced phases of different signs, displacing the interferometric pattern accordingly. The Sagnac-induced phase shift can also be treated as a movement of a standing wave inside the interferometer. Thus, the positions of nodes and antinodes are fixed if the gyroscope is at rest. When a gyroscope is rotated, the photoreceivers measure the rotation angle by counting the interference fringes running over them. Figure 2 schematically shows the interferometric patterns when the gyroscope at rest and during rotation. It is worth noting that the described phase-based detection of the Sagnac effect is a dominant methodology for passive gyroscopes. In the case of passive interferometers, the recorded pattern is sensitive only to a variation of the angular velocity due to a single traverse in the interferometer by the laser beams. In other words, since the laser beams propagate only once through the interferometer (open-loop), the Sagnac-induced phase shift is not accumulated from one roundtrip to another as happens in active gyroscopes. Therefore, passive gyroscopes, generally, provide lower sensitivity. The phase increment of 2π is proportional to the rotation angle of the interferometer. It is usually from 0.1–0.2” for large perimeters of the order of 4 m to 10–20” for small perimeters of the order of 4 cm. In Section 3.3.2 and Section 4, we discuss and demonstrate a novel approach to record interferograms of ultrashort pulses via time-stretch technique in order to retrieve the relative phase associated with the Sagnac effect.

Alternatively, the interference of two narrow-band frequencies will create a beat frequency, equals to the difference between the oscillation frequencies of the incident radiation. The beat-note can be further accurately retrieved from the RF spectrum of the signal from the photodetector. In laser gyroscopes, initially matching counter-propagating oscillation frequencies will demonstrate beat-note, related linearly to the rotation frequency (Equation (Equation 3)). Generally, this method provides the most accurate measurements of the Sagnac effect [17].

Being a feature of the laser cavity, the optical mode beat note is associated with rotation detection using active laser gyroscopes. However, in 1977, Ezekiel and Balsamo proposed a new architecture of a passive resonator laser gyroscope by introducing a closed-loop optical path in the interferometer [40]. The multi-turn element inside, for example, the Fabry–Perot interferometer, allows frequency splitting of resonance optical modes due to the Sagnac effect. Although passive resonator laser gyroscopes have the potential to achieve performance comparable with active laser gyroscopes, they still underperform their contenders [41].

### 2.2. Measurement Errors of Angular Rotation in Laser Gyroscopes

As discussed above, the lock-in effect limits the lowest observable angular velocity. Apart from it, laser gyroscopes inherit the following errors of determining the angle of rotation:*Gas degradation.* He-Ne lasers generate the most desirable narrow-band radiation for gyroscopic measurements. However, gas lasers suffer from gas leaking from the cavity enclosure, which affects the laser gain and, therefore, the scale factor of the gyroscopes. Moreover, the scale factor is further reduced due to contamination of the laser gas with hydrogen, oxygen, nitrogen and water vapour [42,43]. Besides, gas laser gyroscopes face bias due to gas flow caused by temperature fluctuations or ionic flow due to electric discharge [44]. Gas degradation limits the long-term performance of the gyroscope and requires the usage of a solid-state gain medium.*Variation of the scale factor.* Under the influence of external conditions such as temperature and pressure, the scale factor tends to fluctuate, affecting the gyroscopic measurements. The intracavity conditions, such as instabilities of gain and losses, also affect the scale factor of the gyroscope. In fibre gyroscopes, time-dependent temperature fluctuations lead to travel time variation along the same sections of optical fibre for counter-propagating beams (known as the Shupe effect [45]), causing an additional parasitic difference in their optical paths.*Intracavity non-reciprocal effects,* the most significant among which are Magneto-optic effects [46,47] and Kerr effect. In fibre gyroscopes, the minimum detectable angular rate is usually limited by the Kerr effect. Owing to the very small silica fibre core and long distances in fibre interferometer, even small inequality in intensity between counter-propagating beams results in additional accumulated Kerr-induced phase shift [48]. This effect could be eliminated in hollow-core fibre gyros [49].*Optical frequency fluctuations.* All free-running lasers experience a variation of the oscillation frequency of the output radiation. One approach to mitigate this limitation is by applying a feedback loop with piezoelectric actuators [50]. For long-term operation, the oscillation frequency could be stabilised via a beat-note with an actively stabilised reference laser as a feedback reference signal [51].*Polarisation instability.* Counter-propagating beams can experience polarisation instability, leading to a variation of the optical paths caused by birefringence in optical fibres. The effect of polarisation instability could be mitigated by introducing a polariser inside the interferometer. The accuracy could also be enhanced by using a polarisation-maintaining fibre or polarising single-mode fibre [52].*Acoustic noise and vibrations.* Optical fibres are quite sensitive to acoustic noises, which are converted to phase noises of the counter-propagating beams through the photo-elastic effect [53]. Vibrations and input shock introduce additional noise and bias offset to the gyroscope performance. However, they could be minimised by a rigid design of the gyroscope.

All mentioned parameters define the bias stability and performance of the laser gyro. The stability of the gyroscopic measurements is usually defined as an Angular Random Walk (ARW) and expressed as an Allan deviation at 1 s integration time with units rad/s/$\sqrt{Hz}$ (Section 3.2). In other words, the ARW estimates the fluctuation of the beat-note frequency under constant angular velocity. In FOGs, the ARW is limited mainly by the intensity noise of the laser generation [54]. Another major characteristic of a gyroscope stability is a bias drift, which is estimated as a variation range from the mean value of the output signal at long-term operation and usually expressed in degrees per hour.

### 2.3. Gyroscope Performance Consideration

Up to date, extremely large cavity laser gyroscopes with the area up to 367 m${}^{2}$[55] and even 834 m${}^{2}$[56] are the best choice for measurements requiring the highest sensitivity and stability [17] (Figure 3). Remarkably, such ring gyroscopes with monolithic design experimentally resolved a rotation rate of $3.5\;\cdot \;10^{-13}$ rad/s (averaged over 1000 s). They detected the Chandler and the annual wobble of the Earth [57] and perturbations of Earth’s rotation [58,59].

Depending on the application, for example, for the use in inertial navigation systems in aircraft, ships and spacecraft [13,60,61,62], the entire laser gyroscope unit should be compact and highly durable in the presence of spurious vibrations, which generally comes at the price of reduced sensitivity. Nevertheless, most of the applications are well satisfied with typical sensitivity of 0.1/h and 0.001/h–0.0001/h for spacecraft and submarines, respectively (Figure 3).

In 2019, Liu et al. demonstrated improved results on the passive resonant gyroscope [59]. They achieved the sensitivity of 2$\;\cdot \;10^{-9}$ rad/s/$\sqrt{Hz}$ within 1 × 1 m${}^{2}$ interferometer utilising a diode laser locked to adjacent longitudinal modes of a ring cavity. This performance is comparable to active laser gyroscopes of similar size [63].

The sensitivity of the most prominent FOGs can reach $10^{-8}$ –$10^{-9}\;rad/s/\sqrt{Hz}$, which are proposed for use as a portable and reliable gyro for research in seismology [64]. In 2020, Li et al. demonstrated a 15 times improvement in the angular random walk down to 0.0052/$\sqrt{h}$ by using a reciprocal modulation-demodulation technique based on a semiconductor laser [65]. By using Kagome fibre, a resonant FOG reached the angular random walk of 0.004/$\sqrt{h}$[66]. Moreover, a substantial reduction of the angular random walk down to 0.58 mdeg/$\sqrt{h}$ was achieved by suppression of RIN using a Faraday rotator mirror [67]. Further increase in fibre gyroscopes performance could be achieved by exploiting temperature-insensitive fibre interferometers [68], based on hollow-core fibres [69].

Another typology of laser gyroscopes, such as resonant micro-optic gyroscopes, gained much attention due to their low cost and chip-scale size [30,70]. Such devices comprise a narrow linewidth laser source and a high-finesse ring microcavity. Later, Mahmoud et al. proposed a gyroscope based on alteration of refractive index caused by the acousto-optic effect. Such acousto-optic gyroscope showed the angular random walk of 60/$\sqrt{h}$[71]. Another approach for gyroscopic detection applies Brillouin scattering in microcavities. Thus, a microcavity with 18 mm diameter has demonstrated the sensitivity of 15/h/$\sqrt{h}$[72]. Later results reveal the capability of a chip-scale ring cavity gyroscope to detect Earth rotation and sinusoidal rotations with amplitude as low as 5 deg/h [73]. As discussed above, the introduction of strong dispersion can provide an enhancement factor to micro-optic gyroscopes up to $∼10^{4}$ [74]. In parity-time-symmetric laser gyroscopes, comprising two paired rings, the enhancement of the Sagnac effect near exceptional points can achieve factor $10^{8}$ according to numerical estimations [75]. However, thus far, the factor of only ∼300 has been confirmed in experimental observations [76]. With the constantly growing research interest, we envisage a high progress and application potential of gyroscopic measurements based on ultrashort pulses in microcavities.

## 3. Ultrafast Lasers

The mode-locking mechanism has been extensively studied after the first realisation in He-Ne laser, generating 2.5 ns pulses, in 1964 [77]. Modern ultrafast laser achieved direct generation of pulses as short as 5 fs in bulk lasers [78] and 24 fs in ultrafast fibre laser [79]. High peak powers, delivered by ultrashort pulses, with the potential to achieve 100 PW [80] are essential for investigations of relativistic effects and extreme high-field physics [81,82,83]. The generation of equidistant frequency lines, frequency equivalent of the pulse train generation, are widely used for spectroscopy [84] and frequency metrology [85,86]. Mode-locked fibre lasers have proved their high stability performance in an optical clock with frequency instability of $10^{-18}$–$10^{-19}$ [87]. Naturally, ultrafast lasers have found a broad application range in sensing, as well as passive and active gyroscopes, which are in the spotlight of the current *review*.

### 3.1. Principles of Mode-Locking Operation

The basis of passive mode-locking has been investigated for four decades [88,89], using in particular so-called fluctuation model [90,91]. Conventional continuous wave (CW) features chaotic intensity fluctuations as a superposition of numerous longitudinal modes with random phases. Synergistic phase locking (synchronisation) of these modes can be realised through nonlinear interactions, resulting in intensity discrimination: higher-amplitude spikes are supported, while low-intensity fluctuations are suppressed. As a result, the CW generation is transformed into a regular train of pulses with a duration, determined by the gain bandwidth, and a repetition period equal to the cavity round trip time [92].

Ultrashort pulse formation, propagation and amplification in fibre laser cavities is governed by the modified nonlinear Schrödinger equation for a temporal pulse envelope *A* [93]:(5)$$\frac{\delta A}{\delta z}=-\frac{i}{2}\left(\beta_{2}+i\frac{g}{\Omega_{g}^{2}}\right)\frac{\delta^{2}A}{\delta t^{2}}+i\gamma {\left|A\right|}^{2}A+\frac{1}{2}\left(g-l\right)A$$
where *g* and *l* are gain and loss of the cavity, *z* is the propagation distance, $\Omega_{g}$ is the bandwidth of the gain, $\beta_{2}$ is the second-order dispersion coefficient and γ is a nonlinear coefficient, determined by operational wavelength λ, effective fibre mode area $A_{eff}$ and nonlinear refractive index of the laser media $n_{2}$, as $\gamma =2\pi n_{2}/\lambda A_{eff}$.

Figure 4b shows schematically an ultrashort pulse train. The red line demonstrates the intensity envelope (typically recorded by photodetectors), while the green line represents the electric-field carrier wave. The pulse-to-pulse phase slip Δϕ arises due to group retardation of the wave packet and known as a carrier-envelope phase. The time separation between the pulses *T* equals the time of propagation along the laser cavity L/c, where *c* is the speed of light and *L* is the optical length of the cavity. In other words, only one pulse is circulating in the laser cavity at the fundamental repetition rate. In the frequency domain, the pulse train presents equidistant narrow lines (longitudinal modes of the laser cavity) and is usually referred to as a frequency comb (Figure 4c). The frequency separation is dedicated by the laser cavity and equals the repetition frequency of the pulses $f_{rep}=c/L=1/T$. The offset of the comb-lines is governed by the carrier-envelope offset (CEO) frequency $f_{CEO}$, which is directly linked to the carrier-envelope phase in the time domain $\Delta \phi =2\pi \;\cdot \;f_{CEO}/f_{rep}$. Frequency comb generation has a significant role in many applications and sensing, enabling fast dual-comb spectroscopy [84] and high-precision metrology [27].

To realise required nonlinear interactions and, therefore, enable mode synchronisation, inherent nonlinear effects of laser cavity or highly nonlinear materials can be applied. The appropriate nonlinear materials feature saturable absorption behaviour, i.e., the incident low-intensity light excites their electrons to upper energy states, and, when the following excitation is restricted according to the Pauli principle, the radiation of high intensity is transmitted. The other critical parameter of saturable absorbers is the relaxation time, which determines the duration of the generated pulses. In these regards, artificial modulators, utilising the nonlinear optical Kerr effect, feature the shortest recovery time of less than 10 fs since it is based on the interaction of the electromagnetic field with the electrons of laser media [92,94]. In fibre lasers, the ultrafast modulation via the nonlinear optical Kerr effect can be realised by nonlinear polarisation evolution (NPE) [95] or propagating through nonlinear optical and amplifying loop mirrors (NOLM and NALM) [96,97].

Among the first materials exhibiting saturable absorption behaviour are dyes [98,99]. Despite the demonstration of sub-100-fs pulse duration, such lasers had low efficiency and elaborate design. Starting from the 1990s, semiconductor saturable absorber mirrors (SESAMs) received widespread development [100]. These III–V group binary and ternary semiconductors with multiple quantum wells structures were grown on a distributed Bragg reflector [100]. Shortly after the first demonstration and until now, SESAMs have been widely used in research and commercial solid-state, semiconductor and fibre laser systems, operating in a broad spectral range spanning from visible to short-wave infrared range [101,102,103,104]. However, SESAMs fabrication is a sophisticated and expensive process. Therefore, the research community remains in a continuous search for a cost-effective material featuring high nonlinearity and ultrafast relaxation time. During the past decade, numerous new materials, generally nanomaterials, have emerged, including carbon nanotubes (CNTs) [105], graphene [106,107], transition metal dichalcogenides (TMDs) [108,109], MXene [110,111], black phosphorous [112,113] and plasmonic metasurfaces [114]. The key advantage of nanomaterial SAs is the flexibility of their implementation into the laser setup. While SESAM is typically operating as a mirror and, therefore, can be inserted into linear laser configuration or using a circulator, nanomaterials can operate in transmission. The most common approaches for implementation is by sandwiching the SA composite between fibre ferrules [107], deposition on side-polished (D-shaped) or tapered fibres [115], ink-jet printing [116] or infilling microfluidic channels by SA solution [117].

The advantages of ultrafast fibre lasers are not limited to compactness and robustness but also include flexibility of generation regimes and tuneability of output parameters, such as pulse duration and profile, which can be tailored to a particular application. The interplay of dispersion and the nonlinearity, gain and losses with the proper selection of fibre parameters can trigger a large variety of operational regimes. The most common operation is the generation of optical solitons [118] when the positive nonlinearity is compensated by anomalous net cavity group velocity dispersion. With the creation of a dispersion map in the laser cavity (e.g., by sequencing sections of normal and anomalous dispersion), the generation of dispersion-managed solitons [119] can be achieved. Dissipative [120] and self-similar pulses (similaritons) [121] can be formed in all-normal dispersion fibre laser cavities.

By designing the fibre laser cavity so that the normal group-velocity dispersion is compensated by negative nonlinearity, a conventional soliton cannot be generated. Induced broadening and chirping in such cavities give rise to so-called dark solitons, which present a localised sharp deep in CW background with a constant amplitude [122].

Regarding sensing, particularly of the optical Sagnac effect and following gyroscope applications, the influence of nonlinear effects and group velocity dispersion becomes even more crucial. Thus, self-phase modulation and self-steepening effect significantly affect the group and phase velocities of ultrashort pulses and, therefore, determines carrier-to-envelope offset frequency stability [123,124,125], which in its turn determines the resolution of following gyroscopic measurements.

### 3.2. Characterisation of Output Radiation in Ultrafast Lasers

Any laser source is characterised by a number of parameters. The full energy description of the pulsed laser is giving by the next equations:(6)$$P_{av.}=\lim_{t\to \infty }\frac{\int_{-\infty }^{+\infty }A^{2}dt}{t}\approx \frac{\int_{0}^{T}A^{2}dt}{T}=\frac{E_{pulse}}{T}\;\left[\mathrm{Watt}\right]$$
(7)$$E_{pulse}=P_{av.}\;\cdot \;T=\frac{P_{av.}}{f_{rep.}}\;\left[\mathrm{Joule}\right]$$
(8)$$P_{peak}=\frac{E_{pulse}}{\tau_{{}_{FWHM}}}\;\left[\mathrm{Watt}\right]$$
where $P_{av.}$ is the average power of the laser generation, *A* is the pulse amplitude, $E_{pulse}$ is the energy contained in the pulse (pulse energy), $P_{peak}$ is the pulse energy over pulse duration (peak power), $\tau_{{}_{FWHM}}$ is the full width at half maximum (FWHM) pulse duration, *T* is the time period (roundtrip time) and $f_{rep.}=1/T$ is the repetition rate.

Currently, the duration of ultrashort pulses cannot be measured directly by even the fastest photodetector due to the limited rise time of 10 ps, and more complicated techniques should be exploited. The most established methods for measurement of ultrashort pulse duration are based on the autocorrelation of the pulses, i.e., the strobing of a pulse with itself [126]. However, the autocorrelators can provide only an intensity profile but not a full-field characterisation of the optical pulse, such as a pulse chirp or a carrier-envelope phase. The pulse chirp could be understood as a time variation of the instantaneous frequency or non-flat spectral phase, accumulated due to group delay dispersion or self-phase modulation [127], which affect the pulse shape and duration. Nowadays, various devices for measurement of the pulse temporal, spectral and phase profiles, based on the self-strobing of the pulse, have been demonstrated and commercially available (FROG [128], GRENOUILLE [129] and SPIDER [130]). For enhancing sensitivity and resolution, autocorrelators employ nonlinear effects, such as second-harmonic generation (most common), third-harmonic generation or dual-photon absorption [131]. In general, autocorrelators provide the best time resolution (less than 10 fs).

Any precision measurements require high stability of the laser generation. Noises in mode-locked lasers are generally characterised as fluctuations in pulse intensity or average power [132] and noises of frequency comb [133], such as variations in the repetition $f_{rep}$ and carrier-envelope offset $f_{ceo}$ frequencies [134,135]. Fluctuations of $f_{rep}$ determine a timing jitter, i.e., a deviation of the pulse envelope position in the time domain from its perfectly periodic position. While the carrier-envelope offset frequency $f_{ceo}$ manifests as a pulse-to-pulse phase slip, its value is crucial in high-field physics [136] and for the determination of the absolute frequencies of the comb lines. The stability and linewidth of frequency comb lines have a crucial impact on many high-precision measurements (e.g., optical atomic clocks and high harmonic generation, to name a few), and gyroscopic measurements are not exception, since the beat-note frequency corresponds to the comb beating between counter-propagating pulses [123].

The timing jitter characterisation is of particular interest for gyroscopic measurements as uncertainty in pulse position converts to uncertainty in angular rotation. In the frequency domain, the timing jitter power spectral density (PSD) can be treated as the phase noise PSD of $f_{rep}$. The most straightforward method to estimate the timing jitter of the pulsed generation is the evaluation of the shape and width of the repetition frequency $f_{rep}$ by recording the RF spectrum using a photodetector signal [137]. Examples of the RF spectrum are shown in Figure 5a. Here, a mode-locked fibre laser operates in two different generation regimes set by adjustable parameters (intracavity polarisation state, pump power, etc.) [138]. As depicted, central peaks corresponding to the fundamental repetition rate are almost identical. However, one of them inherits strong sidebands, which diminish the total SNR by 15 dB. Unfortunately, this method cannot provide timing jitter at low frequencies due to the non-repetitive dynamics of mode-locked lasers [139]. Recent works show jitter measurements with attosecond precision using a simple optical heterodyne technique [140] or balanced optical cross-correlation [141].

Another major characteristic of the frequency stability is the Allan deviation. Allan deviation is a square root of the Allan variance, which can be calculated as:(9)$$\sigma_{y}^{2}\left(\tau \right)=\frac{1}{2}<{\left(\bar{y_{2}}-\bar{y_{2}}\right)}^{2}>$$
where $\sigma_{y}\left(\tau \right)$ is Allan deviation with units $\sqrt{Hz}$ and $\bar{y}$ is the averaged value over the time τ. The Allan deviation is characterised by the difference between two averaged consecutive measurements, which make it distinctive from the traditional standard deviation from the mean value. In mode-locked lasers, Allan deviation is usually assessed at fundamental repetition frequency $f_{rep}$. Generally, Allan deviation estimates the ability of the measured frequency to remain unchanged. The example of Allan deviations of free-running mode-locked fibre laser repetition rate at two different generation regimes is depicted in Figure 5b [138]. Nowadays, the most prominent laser systems with active stabilisation achieve an uncertainty at the level of $10^{-15}$ at 1 s time scale [142,143]. Generally, the instability of repetition frequency in time scales below ∼0.4 s is governed by the laser system stability. At the same time, the stability starts to diverge at the time scales above ∼0.4 s due to temperature fluctuations [142,144]. For laser gyroscopes, Allan deviation is measured for the beat-note frequency and defines the angle random walk (ARW).

The carrier-envelope offset frequency characterises the phase stability of the pulse train and greatly affects the overall performance of the generated frequency comb. The CEO frequency can strongly fluctuate in the presence of intensity noises of pump source [145] and due to environmental perturbation [124]. Moreover, the stability of the CEO frequency depends on the generation regime of ultrashort pulses [133]. Thus, the CEO frequency fluctuations can be substantially suppressed in a laser with close-to-zero intracavity dispersion [146]. However, the CEO frequency performance is limited by amplified spontaneous emission in the laser cavity [147]. The non-trivial task of CEO phase characterisation was presented for the first time in 1996 by Xu et al. [148]. However, the demonstrated method provided only relative changes between consecutive pulses. The absolute CEO frequency measurements employ an f−2f interferometer, based on at least one-octave supercontinuum generation and second harmonics generation [149,150]. Another technique is based on the ionisation of an isotropic medium [151]. However, both methods for absolute measurements of $f_{ceo}$ require complex setup and high input energies. In bidirectional mode-locked lasers with identical repetition rates of counter-propagating generation, CEO frequency difference can be easily estimated from the RF spectrum [152].

Stabilisation of the mode-locked lasers in the frequency domain with sub-hertz linewidth can be achieved by controlling the two degrees of freedom $f_{ceo}$ and $f_{rep}$ with high-speed actuators [153]. While the $f_{ceo}$ frequency is stabilised through feedback control of the pump laser diode, the $f_{rep}$ is usually stabilised by locking to a reference continuous-wave laser with intra-cavity stabilisation [154].

The intensity noise of both continuous-wave and pulsed generation is characterised as the relative intensity noise (RIN). Basically, RIN is a ratio of the average mean-square fluctuation of the optical power $\langle \delta P{\left(t\right)}^{2}\rangle$ to the square of the average optical power ${\langle P\left(t\right)\rangle }^{2}$ over different acquisition times [137,155,156]. In the Fourier frequency domain, the RIN described as:(10)$$S_{RIN}\left(f\right)=\frac{2}{\bar{P^{2}}}\int_{-\infty }^{+\infty }\langle \delta P\left(t\right)\delta P\left(t+\tau \right)\rangle exp\left(i2\pi f\tau \right)d\tau \;\left[{\mathrm{Hz}}^{-1}\right]$$
where $\bar{P}$ is average optical power. The RIN PSD can be derived from the RF spectrum analyser. Figure 5c demonstrates the RIN PSDs for two operation regimes in a free-running ultrafast fibre laser [138]. Paschotta [157] studied the influence of the intensity noise on timing jitter and indicated the following coupling mechanisms: slow saturable absorber, Kramers–Krönig-related phase change, nonlinear Kerr effect and Raman self-frequency shift. The intensity noise of free-running mode-locked fibre lasers is limited by the RIN of the pump source. Thus, the power fluctuations of the mode-locked lasers could be efficiently stabilised by electronic feedback control of the current of the pump laser diode [158,159] or with an acousto-optic modulator [160].

### 3.3. Real-Time Measurements of Ultrashort Pulse Dynamic

The above-described methods are used to measure time-averaged characteristics of mode-locked lasers and, therefore, are effective when quantifying laser long-term operation stability. However, mode-locked fibre lasers are known for complex nonlinear dynamics owing to the multidimensional parameter space of their operational regimes [161]. Notably, the typical time scales of nonlinear dynamics are in the cavity round trip time order, which is well beyond the bandwidth of the conventionally used methods described above. Observation of these fast time-scale dynamics becomes crucial if one needs to understand the ultrafast nonlinear events in these lasers. Therefore, real-time measurements of fibre laser dynamics have been recently established for observing the nonlinear dynamics at the high temporal and spectral resolution, preferably over long times scales. In this section, we highlight two of the commonly used methods of intensity and spectral-domain characterisation.

#### 3.3.1. Intensity Domain Methods

Real-time observation of intensity domain dynamics is challenging because their accurate capture needs a high temporal resolution comparable to the theoretical transform-limited pulse width of the radiation and observation over long time scales. Currently, the most advanced combinations of commercially available photodetectors and digital storage oscilloscopes offer the resolution of a few tens of picoseconds. At the same time, the possibility of recording long windows of data of tens of giga-samples allows continuous real-time observation of fibre laser dynamics over the time scales of hundreds or even thousands of round trips. One can then segment this long record into equivalent round-trip length segments, which can be stacked in a two-dimensional matrix form to acquire a spatiotemporal representation of the laser dynamics [162]. Such 2D maps help to reveal how the laser generation evolves within consecutive round trips. Figure 6 demonstrates the principle of spatiotemporal dynamics of periodic pulses in bidirectional mode-locked fibre laser [29]. The temporal resolution is limited by the oscilloscope-detector combination according to the following expression [163]:(11)$$t_{FWHM}=\sqrt{{{t_{FWHM}}^{2}}_{PD}+{{t_{FWHM}}^{2}}_{DSO}}$$
where ${t_{FWHM}}_{PD}$ and ${t_{FWHM}}_{DSO}$ are the time-domain impulse response of photodetector and oscilloscope, correspondingly, which relate to the rise time $t_{R}$ as $t_{FWHM}$ = 0.915$\;\cdot \;t_{R}$. The temporal resolution limits of the above method can be surpassed using nonlinear approaches. One such approach is exploiting space-time duality to realise a time lens [164,165], wherein a pulse can be stretched without changing its temporal profile by imprinting a quadratic phase on it.

This effect has been used for real-time full-field characterisation of soliton dynamics with a temporal magnification of 76.4, which results in 400 fs resolution [166].

The spatiotemporal technique has been effectively used for observing the laminar-turbulent transitions [167], interaction of quasi-stationary localised structures [162] and rogue events [168]. More recently, the method has been used for capturing the onset dynamics of pulsed laser generation in Ti:Sapphire lasers [169] and unidirectional [170] and bidirectional fibre lasers [171,172]. The primary advantage of this method is that it is a linear approach to capturing fast-evolving dynamics, resulting in a simple measurement configuration.

#### 3.3.2. Spectral Domain Methods

The spectral measurement of fibre lasers using conventional optical spectrum analysers provides valuable information about their operational regime when complemented with the time domain and RF measurements. Optical spectrum analysers are typically based on a diffraction grating, which scans over a specified wavelength span, and photodetector, measuring spectrally resolved power. Therefore, the spectra obtained using these devices are again highly averaged with the maximum data acquisition speeds in the order of a few Hz. One of the elegant and most widely used methods for studying round trip-resolved pulse spectra is the Dispersive Fourier Transform (DFT) [173]. Similar to the time lens technique, the DFT exploits the space-time duality. Here, a pulse is sent through a long dispersive line, allowing it to disperse linearly in the temporal domain analogous to the free-space spatial propagation of aperture-limited optical radiation over a long distance. In the spatial domain, this leads to the formation of the Fraunhofer diffraction pattern of the aperture. In the temporal domain, this reconstructs the pulse spectra. Figure 7a presents a conceptual setup for the DFT measurements with optical fibre as a dispersive element. The experimental single-shot DFT measurements of two-pulse interference are depicted in Figure 7b. The spectral resolution can be increased by accumulating larger chromatic dispersion, e.g., during propagation through a longer optical fibre section or diffraction gratings with a larger dispersion. The broadening of the pulse in time domain Δτ can be estimated for such case as:(12)$$\Delta \tau =Dz\Delta \lambda +\frac{1}{2}\frac{dD}{d\lambda }z{\left(\Delta \lambda \right)}^{2}$$
where *D*, dD/dλ and *z* are dispersion, dispersion slope and the length of optical fibre, correspondingly. Similar to spatiotemporal analysis, the final resolution of the measurements is determined by the bandwidth of the used oscilloscope and high-speed photodetectors.

The mathematical properties of the Fourier transform can be further exploited to obtain the pulse-resolved first-order autocorrelation function (ACF). Owing to the Wiener–Khinchin theorem, the single-shot ACF could be obtained by applying the fast Fourier transformation of individual single-shot (of one round trip) pulse spectrum. The resolution of the retrieved ACF is inversely proportional to the covered wavelength span and can achieve a few tens of fs. The maximum delay time is determined by the DFT resolution and usually is limited from tens to hundreds of picoseconds. Real-time ACF attains particular interest for studying dynamics of complex coherent structures, e.g., multiple solitons. Numerically, in the presence of two-pulse intrusion, the ACF expressed as [161]:(13)$$R\left(\tau^{\prime }\right)=2\int I_{0}\left(\omega \right)e^{i\omega \tau^{\prime }}d\omega +e^{-i\phi }\int I_{0}\left(\omega \right)e^{i\omega (\tau +\tau^{\prime })}d\omega +e^{i\phi }\int I_{0}\left(\omega \right)e^{i\omega (\tau -\tau^{\prime })}d\omega$$
where $I_{0}\left(\omega \right)$ is a spectral intensity of a single pulse, τ is time separation between pulses and ϕ is a relative phase. An example of the experimental results of two closely-separated pulses is shown in Figure 7c. By extraction of the relative phase, internal dynamics of soliton molecules have been experimentally revealed [174]. As discussed below, the encoded phase data can also be used for sensing application, e.g., gyroscopic measurements based on Sagnac phase shift.

Apart from a fundamental interest in nonlinear fibre optics, several sensing applications, which require high acquisition rates, have been demonstrated, based on the DFT measurement, e.g., displacement sensing and barcode reading [175], cellular imaging [176], imaging of ultrafast laser ablation [177] or microfluidic flow imaging [178]. Recently, a spectrally scanning LiDAR system with MHz line rate has been presented [179]. However, this field is still unexplored, and we expect more sensing application based on the DFT measurements to emerge in the near future.

### 3.4. Mode-Locked Fibre Laser Designs for Gyroscopy

First pulsed fibre laser gyroscopes featured laser cavities formed by conventional broadband mirror and a fibre-optic loop mirror [180]. The loop mirror generally comprises a directional coupler and a coiled fibre section (Figure 8a). A signal is applied to phase modulator to adjust the phase difference between counter-propagating pulses in the loop. When the signal has a frequency equal to the longitudinal mode spacing of the laser cavity, mode-locking occurs. It is a loop mirror that played the role of rotation sensitive element. Thus, due to the rotation of the loop mirror, the Sagnac phase shift accumulates between two pulses propagating along the fibre coil, which is linearly proportional to the rotation rate.

Similar to established free-space laser gyroscopes, the ring fibre laser configurations (Figure 8b,c) presents another approach, which recently has attracted considerable research attention. However, the realisation of mode-locked generation in opposite directions in a ring cavity imposes several fundamental restrictions. Thus, the cavity should be designed in such a way that saturable absorber upon pulse arrival of both pulses should be in the ground state. Furthermore, the application of artificial modulators based on nonlinear polarisation evolution is restricted. In the lasers mode-locked solely by nonlinear polarisation evolution, the bidirectional operation could not be obtained only above the mode-locking threshold. As soon as the conditions for reaching mode-locking operation are fulfilled, one of the directions becomes dominant over another with around 10-dB extinction ratio, leading to quasi-unidirectional generation [181,182]. Therefore, the simplest ring cavity design uses material saturable absorbers (Figure 8b). When the repetition rate of counter-propagating pulses are synchronised, the pulses typically overlap at the saturable absorber, realising so-called colliding mode-locking [183]. Colliding mode-locking occurs when saturable absorbers operate in transmission, such as various nanomaterials (carbon nanotubes, graphene, MXENE, TMDs, etc.). The benefit of colliding mode-locking is reduced pulse operation threshold, as both counter-propagating beams saturate the absorber.

In 1984, the use of a co-action of two SAs was proposed to increase the performance of mode-locked lasers [184]. Nowadays, many mode-locking fibre lasers are working with co-action of artificial modulators, based on nonlinear Kerr effect and material saturable absorber [185,186]. Co-action of two saturable absorbers allows overcoming their individual disadvantages and increase the laser performance. In such a case, while one saturable absorber works at the lower energies and supports the mode-locking start-up, another modulator shortens the pulse duration due to a shorter relaxation time.

Nevertheless, nonlinear interaction between counter-propagating pulses in a common saturable absorber can cause undesirable instability of the pulses, such as a significant phase noise. Moreover, when the counter-propagating beams share exactly the same optical path inside the laser cavity, it restricts the active stabilisation of the output radiation, since stabilisation in one direction can introduce instability in the counter-propagation beam. One of the ways to overcome this bottleneck is to separate both beams in different fibre and use two different saturable absorbers. Thus, few works suggested the application of one or two fibre-based optical circulators to separate counter-propagating pulses and enable two mode-locking mechanisms, typically, using two SESAMs (Figure 8b) [187,188]. Such scheme also allows controlling the repetition rate of counter-propagating pulses individually by adjusting the length of circulator ports. The obvious drawback of this setup is a higher cost due to the implementation of extra components. Some compromise can be achieved by using laser cavity geometry, as shown in Figure 8c. Here, the setup comprises two optical circulators and only one SESAM. However, optical circulators also restrict the laser performance by limiting operational bandwidth and, together with optical couplers and isolators in the cavity introduce substantial losses, lowering the laser efficiency.

Finally, over the last decade, several works have demonstrated a wide variety of bidirectional lasers operation wavelength, such as at 1.55 and 2 μm in Tm-doped fibre lasers [189,190,191], and mode-locked pulse generation regimes by varying net cavity dispersion [192,193,194]. However, according to the Sagnac Equations (Equation 3) and (Equation 4), the application of shorter operation wavelengths is more beneficial for increasing the overall sensitivity to rotation.

## 4. Highlights of Ultrashort Pulsed Fibre Laser Gyroscopes

The first results on gyroscopic measurements in ultrafast fibre lasers were presented in 1993 [25]. The gyroscope laser cavity, similar to the one presented in Figure 8a, comprised Neodymium-doped double-clad fibre pumped via diode array and a Sagnac interferometer, based on a fibre loop of 600 m length and coiled in a spool with the radius of 8 cm. The mode-locked generation was achieved via active phase modulation at the frequency corresponding to the cavity longitudinal modes spacing. The Sagnac effect was evaluated through the temporal shift. The lowest demonstrated measured rotation rate was 5 deg/s. Although no resolution was stated in this first study, this approach showed potential for increasing acquisition rates for gyroscopic measurements beyond the kHz range. Moreover, this work demonstrated that the Sagnac effect could be enhanced by managing the amplitude of the phase modulator.

Further investigations revealed that optical fibre birefringence of both the loop and the linear part in the gyroscope affects the properties of nonreciprocal pulses and, therefore, output pulse patterns. Nevertheless, nonreciprocal pulses can be controlled by adjusting intracavity polarisation [195]. Thus, in 1997, the sensitivity and angular random walk of mode-locked fibre laser gyroscope were advanced up to 0.06/$\sqrt{h}$ by using a polarisation-maintaining fibre with intracavity polariser [196].

Importantly, Hong et al. [180] demonstrated real-time gyroscopic measurements of dynamic rotation input. The performance of the time-based measurements of the Sagnac effect was increased by implementing a lock-in amplifier, which improved the long-term stability of the bias drift down to 20/h. We want to emphasise that, although the gyroscope was based on the mode-locked laser, during the measurements, the Sagnac effect was not accumulated from each round trip, which significantly limited the gyroscope performance.

Later, Braga et al. demonstrated a beat-note response to applied phase modulation of a bidirectional mode-locked fibre laser [197]. Although the beat-note response was not associated with the Sagnac effect, the phase shift caused by a modulator is similar to the one experienced by laser beams under rotation exposure. Therefore, this result contributes directly to the development of the measurement methodology of ultrashort pulsed laser gyroscopes. In this work, the mode-locked generation was achieved via nonlinear polarisation evolution combined with two amplitude modulators, used to control the crossing point of counter-propagating pulses inside the laser cavity. It is worth noting that the control of the crossing point is essential for further mitigation of the lock-in effect. Moreover, amplitude modulators allowed the repetition rate control with demonstrated uncertainty of ±100 Hz. Unfortunately, the implementation of intracavity phase modulators compromises the integrity and, therefore, benefits the all-fibre laser gyroscope configuration.

### 4.1. Beat-Note Measurements of the Optical Sagnac Effect

The first genuinely Sagnac frequency measurements in ultrafast fibre laser were presented by Krylov et al. in 2017, almost 20 years after the first demonstration of mode-locked laser gyroscope [123]. The laser gyroscope setup was built in a configuration similar to the one shown in Figure 8b with an Er-doped fibre as a gain media and the net anomalous dispersion of −0.11 ps${}^{2}$[198]. The generation of almost bandwidth-limited Gaussian pulses in both directions was enabled by the above-discussed hybrid mode-locking mechanism, combining nonlinear polarisation evolution and carbon nanotubes material saturable absorber. The nonlinear polarisation evolution was realised by implementing coiled polarising fibre and a pair of polarisation controllers. Generally, polarisation selective components in the cavity allow changing the intensity distribution of the field components along the two principal polarisation axes and, therefore, the threshold of intracavity nonlinear effects. Thus, in this work, intracavity polarisation state control and pump power variation allowed fine-tuning of the beat-note frequency, which corresponds to zero rotation [123]. This ‘rest’ beat-note (at zero rotation) refers to the beating of CEO frequencies of counter-propagating pulses [152]. Therefore, the fine-tuning of the ‘rest’ beat-note could be attributed to the CEO frequencies variation of both counter-propagating pulses. This feature opens a possibility for complete elimination of the ‘lock-in’ effect by setting the ‘rest’ beat-note sufficiently apart. However, the sensitivity to the CEO frequency during gyroscopic measurements has a deleterious effect on the bias stability, since the CEO frequency experience rapid fluctuations. Nonetheless, the CEO frequency could be independently measured by f−2f interferometer and efficiently stabilised. Moreover, as discussed above, the fluctuation of the CEO frequency could be minimised by constructing the laser cavity with close-to-zero net cavity dispersion [133].

The results of gyroscopic measurements in the above-described mode-locked laser setup is presented in Figure 9. The ring laser gyroscope presented a single fibre coil with a diameter of ∼1 m and a total area of 0.79 m${}^{2}$. The RF spectrum is shown in Figure 9a with the ‘rest’ beat-note frequency set to 1.61 MHz. The observed beat-note peak had a remarkably high signal-to-noise ratio of ∼30 dB. In the time domain, the pulse train exhibited a harmonic beat-note pattern with nearly 100% modulation depth, as shown in Figure 9b. The gyroscopic effect manifested through the beat-note frequency shift in regard to the ‘rest’ frequency due to the opposite shifts of CW and CCW combs in the frequency domain. The angular velocity ranged from 0.12 to 90 deg/s. As expected, the relation between the rotation rate and the beat-note frequency shift is almost linear (Figure 9c). The measurement resolution reached 0.01/s, estimated by the bias frequency drift. The limitation of the sensitivity was mostly attributed to the carrier-to-envelope offset frequency stability [124].

### 4.2. Beat-Note Measurements of the Optical Sagnac Effect Employing Dark-Soliton Fibre Laser

In 2019, an alternative ultrafast laser operation regime was used for angular rotation measurements using the same beat-note frequency analysis methodology. Unlike conventional ‘bright’ pulses, the presented mode-locked laser gyroscope generated ‘dark’ solitons [199]. Dark solitons present a train of rapid intensity dips in a continuous wave laser radiation with a duration in the order of pico- or nanoseconds [200]. Figure 10a demonstrates a train of dark solitons. Dark solitons have been numerically proved to be less sensitive to noise [201], fibre losses [202] and interactions between neighbouring dark solitons [203], compared to the bright solitons. Additionally, there is no need to implement a delay line to overlap counter-propagating dark solitons in time. Thus, the generation of dark soliton can combine the advantages of both conventional solitons and continuous-wave radiation.

In [199], the ring laser cavity was comprised of two rare-earth-doped optical fibres. An Erbium-doped fibre was used as an active media, while a Thulium-doped fibre with strong absorption at 1550 nm acted as a saturable absorber. The ‘rest’ beat-note frequency was close to one observed in the above-discussed bright soliton laser gyroscope, of 1.4 MHz shown in Figure 10b. The linewidth of the beat-note peak was 169 Hz with a signal-to-noise ratio of ∼7.5 dB. The gyroscope scale factor constituted 3.31 ± 0.09 kHz/(deg/s), which corresponds to a resolution of gyroscopic measurement of 0.05/s. The linear dependence of beat-note frequency on rotation rate is presented in Figure 10c. These results demonstrate that gyroscopic measurements are not limited to conventional solitons, while other generation regimes could be more beneficial for gyroscopic applications.

### 4.3. Real-Time Measurements of the Optical Sagnac Effect

As discussed in the sections above, real-time measurements have provided new insights into ultrafast laser dynamics in both the intensity and spectral domains. Therefore, with the rapid development of these techniques, their applications are were significantly extended, including gyroscopic measurements [29] capable of interrogating the optical Sagnac effect in real-time. In [29], the laser cavity was similar to one shown by Krylov et al. [123]. A distinctive feature of the laser was a possibility to control generation directivity and repetition rates of counter-propagating pulses by adjusting intracavity polarisation state [29,204]. Thus, the bidirectional operation regime could be achieved at slightly different repetition rates or synchronisation. The difference in repetition rate could be easily controlled via polarisation controllers in a region of 50–100 Hz with a central peak at 14.78 MHz and SNR of ∼60 dB. The optical Sagnac effect in laser operating in these regimes was evaluated using two real-time techniques, spatiotemporal intensity and DFT, correspondingly. The timing jitter, evaluated from the RF spectrum, was assessed to be ∼0.87 ps, which guaranteed the stability of the real-time measurements. The gyroscopic measurements laser setup was coiled on a rotating platform with a diameter of 0.63 m and a total area of 0.3 m${}^{2}$.

#### Spatiotemporal Measurements

First, the bidirectional laser gyroscope generation regime with unsynchronised repetition frequencies of counter-propagating pulses was analysed employing spatiotemporal intensity dynamics. Unlike previously discussed works [25] on temporal measurements of ultrashort pulses experienced Sagnac timing shift (Equation (Equation 2)), in [29], the Sagnac timing shift was accumulated from each roundtrip. By using a 33 GHz oscilloscope and a 50 GHz photodiode with a rise time of 12.5 ps, the temporal resolution reached 25 ps (Equation (Equation 11)). Due to the difference in repetition rates of counter-propagating pulses, their crossing point was moving along the laser cavity. Therefore, the recorded spatiotemporal evolution of counter-propagating pulses at the output coupler presents two crossing line trajectories (Figure 11a). The memory of the digital storage oscilloscope was sufficient to capture several round trips of the crossing points. The crossing point of pulse trajectories corresponds to pulse overlap at the output coupler. The gyroscopic effect manifested through the change in the angle between pulse trajectories. The Sagnac effect was evaluated by comparing the temporal separation between counter-propagated pulses after 10${}^{4}$ roundtrips after the trajectories crossing point in regards to pulse spacing when platform in rest (Figure 11b). Figure 11c shows the relation between angular rate and accumulate experimental scale factor of 0.885 deg/s/ns. The experimentally achieved resolution was 0.022/s. Since Figure 11b, the resolution could be further increased by acquisition over a larger number of roundtrip. The maximum roundtrips number and, therefore, the resolution were limited by oscilloscope memory with a maximum number of roundtrips of 3.96 · 10${}^{5}$ for the demonstrated laser cavity.

The same technique also demonstrated a high potential to be used for angular velocities monitoring at MHz data acquisition rates, which are equivalent to the round trip time of the laser. Naturally, an increase of acquisition rate arrives at the expense of the decreased measurement resolution by the same factor. The suitable trade-off could be determined according to the demands of the final application. Thus, for example, a moderate resolution of the Sagnac effect of 94 mrad/s can be achieve with an acquisition rate as high as 418 kHz. These results surpass current state-of-the-art commercially available modalities by at least two orders of magnitude.

#### Dispersive Fourier Transform Measurements

The most noticeable results of real-time angular rotation measurements were achieved by using DFT. This approach requires synchronised repetition rates in both directions in order to record the pulse-to-pulse evolution of the interferometric pattern (Figure 12a).

The lengths of output arms were adjusted for counter-propagating pulses to achieve the time separation between pulses of ∼100 ps. Pulses trajectories are presented in Figure 12a. Therefore, propagating through the DFT line, realised using 11 km of dispersion-compensating fibre with total accumulated dispersion of −1200 ps/nm, the pulse spectra would overlap, producing an interference pattern (as shown in Figure 7). The bandwidth-limited resolution of the DFT measurement using a 33 GHz oscilloscope and a 50 GHz photodiode was 0.018 nm. Figure 12b shows the recorded interference pattern featuring a pulse-to-pulse phase shift, which relates to the carrier offset phase, accumulating over roundtrips. The Sagnac effect manifested as an additional phase shift, changing the tilt of the interference pattern. The angle was extracted by applying the fast Fourier transformation of DFT spectra over a window of 5000 consecutive round trips. Figure 12c demonstrates the FFT frequency difference for the gyroscope platform rotation at two angular velocities. The measurement resolution reached 7.2 mdeg/s. The experimentally obtained scale factor, with relation shown in Figure 12d, was 17.2 mdeg/s/kHz. Additionally, similar to spatiotemporal methods, the resolution could be further increased at the cost of the acquisition rate, i.e., by taking FFT of a larger dataset. Additionally, these single-shot measurements have a potential to overcome the Shupe effect, since the data rates were much higher than the thermal effects. This phase-based technique could also be extended for applications in passive ultrafast laser gyroscopes.

### 4.4. Rotation Sensing by All-Fibre Bidirectional Optical Parametric Oscillator

The above-discussed bidirectional mode-locked fibre lasers feature complex nonlinear dynamics, involving interactions in gain media and collisions in material saturable absorbers [171,172]. A quite different approach for fibre laser cavity design was suggested by Gowda et al. [152], which eliminates the complex dynamics in a gain medium that generally has a relatively long excited state lifetime. The gyroscope in this case presents an optical parametric oscillator (OPO) based on four-wave mixing in an optical fibre loop (Figure 13a). The seed signal with 5-mW average power is split into halves using a variable coupler to obtain the idler signal at around 1620 nm (see inset in Figure 13a). The ring parametric oscillator comprises polarisation-maintaining dispersion compensation fibre and standard SMF-28 with overall dispersion of −0.014 ps${}^{2}$. The OPO loop is designed in such a way to achieve temporal overlap of counter-propagating idler pulses in the variable coupler to simplify the beat-note detection.

The RF spectrum of the OPO output demonstrates a fundamental repetition rate peak and two symmetrical signals due to the difference of the CEO frequencies of the two frequency combs. The rotation of the gyroscopic laser system causes the shift of the beat-note peaks. Therefore, the full width at half maximum of the beat-notes determines the sensitivity of rotation measurements. While the bandwidth of used RF analyser allowed only 300 Hz resolution of the linewidth of $\Delta f_{CEO}$ peaks, time-domain measurements of the voltage from a slow photodiode using an AC-coupled digital oscilloscope allowed improving the resolution down to ∼5 Hz. Figure 13c demonstrates a characteristic sinusoidal signal from the oscilloscope screen with corresponding frequency obtained using FFT, as an inset. The resolution of the $\Delta f_{CEO}$ linewidth could potentially be improved by analysing longer time span. However, in the demonstrated experimental settings, the central frequency of the beat-note drifted by tens of kHz over 30 s due to the environmental perturbations, including temperature fluctuations, or the splitting ratio alteration of the variable coupler. Even though Gowda et al. [152] did not demonstrate rotation measurements, the presented gyroscope concept has high potential to reach high measurement sensitivity.

## 5. Conclusions and Perspectives

Laser technology development underwent a big step forward during the last couple of decades. The evolution from relatively simple design to complex integrated setups with electronics, multiple feedback sensors and machine learning will undoubtedly continue in the future, accelerated by existing and newly emerging applications. Since ultrafast lasers have been comprehensively investigated, the particular interest moves research forward to develop new ultrafast applications with a real-time feedback. Thus, most recently, mode-locked fibre lasers have shown themselves as a perspective tool for optical Sagnac effect detection. Even the first few works have confirmed such advantages of mode-locked fibre laser gyroscopes over the continuous-wave operation as a much higher data frequency and the lack of the dead-band. The rising interest in inertial guidance systems has created a demand for reliable and compact gyroscopes with a good correspondence between temporal resolution and the accuracy of angular velocity measurements. The bidirectional regime of operation in mode-locked lasers can occupy a niche of gyroscopic measurements and satisfy high requirements. Nevertheless, there is still a big room for further ultrafast laser gyroscope modification, generation regime optimisation and enhancement of angular measurement resolution, which motivates both fundamental investigation and industrial development. In this vein, recently demonstrated novel methods to detect the Sagnac effect by using spatiotemporal intensity processing and dispersive Fourier transformation provide a new view on the gyroscopic measurements. We summarise current achievements in the field in Table 1 underlining benefits and bottlenecks of each of realisations.

We can foresee that future research activities in the field will be focused on accelerating data acquisition rates, e.g., using new real-time measurements and laser generation stabilisation. For the latter, new tailored-made hollow-core optical fibres [49] and novel laser configurations could be of particular interest, such as the application of unidirectional laser operation [152,205], which is free from complex nonlinear ultrashort pulse interaction dynamics. Application of innovative machine learning approaches could provide stabilisation of the generation regime by providing controllable feedback [206,207], as well as improved resolution when applied at the data post-processing stage. Overall, ultrafast fibre laser development offers big room for engineering innovations, providing a wide range of possibilities for significant breakthroughs in rotation sensing.

## Figures and Tables

**Figure 1 sensors-21-03530-f001:**
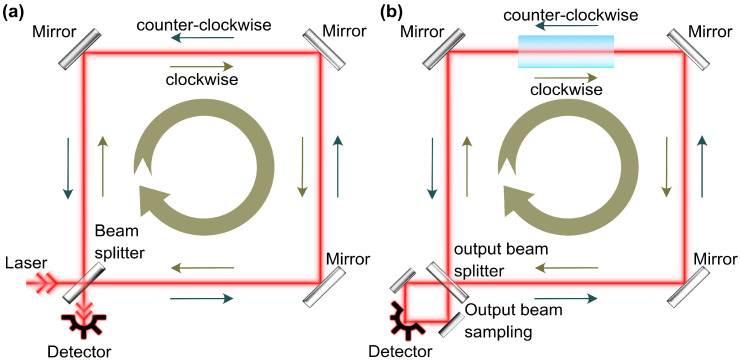
Schematic representation of: (**a**) passive Sagnac interferometer; and (**b**) ring laser gyroscope.

**Figure 2 sensors-21-03530-f002:**
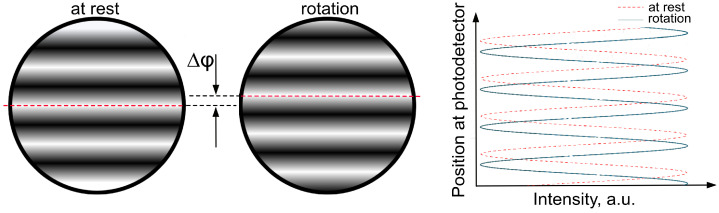
The interference of counter-propagation pulse on photodetector measured from gyroscope at rest and in rotation.

**Figure 3 sensors-21-03530-f003:**
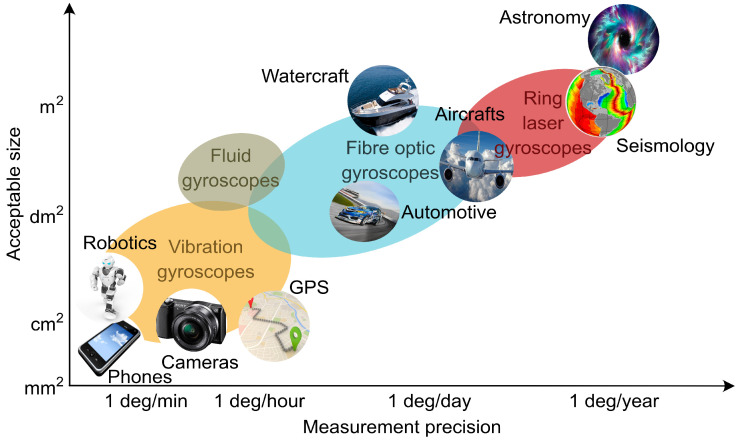
Map of the performance capabilities of state-of-the-art gyroscopes.

**Figure 4 sensors-21-03530-f004:**
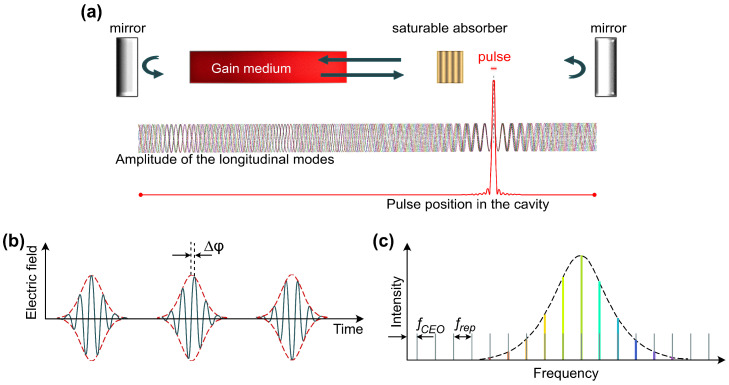
(**a**) Schematic demonstration of the phase locking of multiple longitudinal wave modes. Ultrashort pulse representation in: (**b**) time domain; and (**c**) frequency domain.

**Figure 5 sensors-21-03530-f005:**
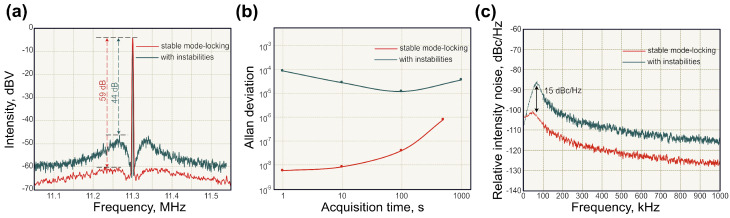
Characteristic of ultrafast lasers: (**a**) radio frequency spectrum at the fundamental repetition rate; (**b**) Allan deviation of fundamental rate; and (**c**) relative intensity noise for soliton operation regimes in a free-running fibre laser with total intracavity dispersion of −0.005 (red line) and −0.021 (blue line). Adapted with permission from [138] *©* The Optical Society.

**Figure 6 sensors-21-03530-f006:**
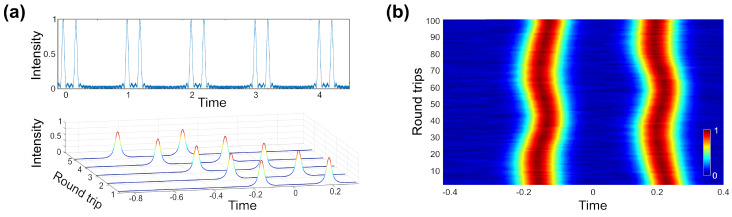
Principles of real-time measurements of spatiotemporal dynamics: (**a**) periodic pulses recorded by the oscilloscope are converted to 2D evolution of selected pulse pairs separated by the round trip time; and (**b**) example of spatiotemporal dynamics of two pulses. Adapted from [29] (*©* CC BY licence).

**Figure 7 sensors-21-03530-f007:**
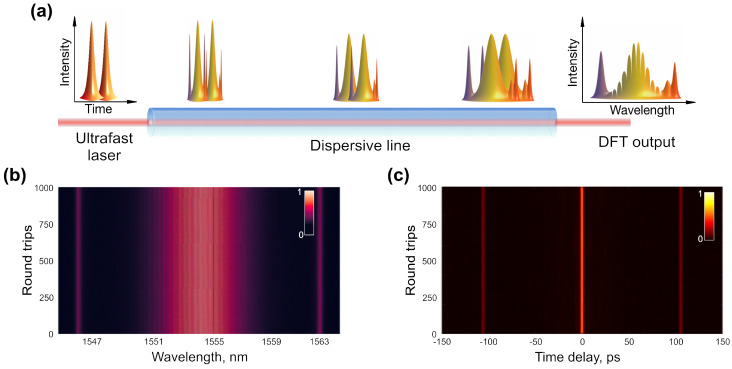
Principles of real-time measurements using the dispersive Fourier transform technique: (**a**) a train of closely-separated pair of optical solitons while propagating through highly dispersive media, mapping their temporal profile to spectral; (**b**) example of recorded single-shot spectral evolution; and (**c**) real-time evolution of field autocorrelation, obtained by fast Fourier transform of single-shot spectra. Adapted from [29] (*©* CC BY licence).

**Figure 8 sensors-21-03530-f008:**
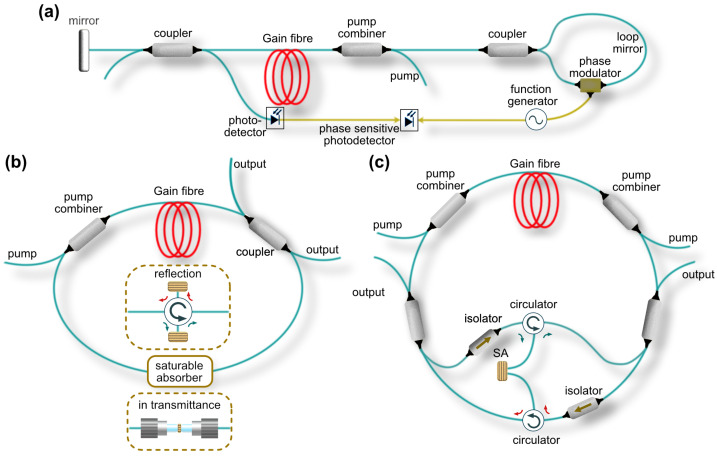
Ultrafast fibre laser cavity designs for gyroscopic applications: (**a**) laser cavity based on nonlinear fibre loop, controlled by phase modulator; (**b**) bidirectional ring all-fibre cavity with possible approaches for saturable absorbers implementations, operating in transmission or reflection; and (**c**) ring cavity setup with separated optical paths for counter-propagating pulses at saturable absorber.

**Figure 9 sensors-21-03530-f009:**
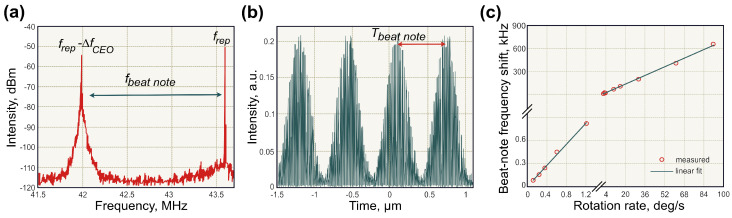
Optical Sagnac effect detection using beat-note measurement: (**a**) RF spectrum and (**b**) oscilloscope trace of combined counter-propagating pulses; and (**c**) rotation measurements at slow and fast angular velocities, demonstrating the scale factor of 6.95 kHz/(deg/s). Adapted with permission from [123] *©* The Optical Society.

**Figure 10 sensors-21-03530-f010:**
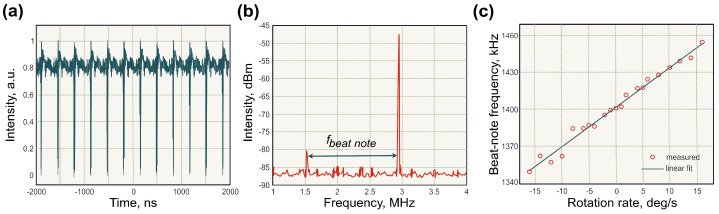
Measurement of optical Sagnac effect using beat-note measurement in bidirectional dark soliton fibre laser: (**a**) typical pulse trace of dark solitons; (**b**) RF spectrum of combined counter-propagating pulses; and (**c**) rotation measurement demonstrating sensitivity of 3.31 kHz/(deg/s). Adapted with permission from [199] *©* The Optical Society.

**Figure 11 sensors-21-03530-f011:**
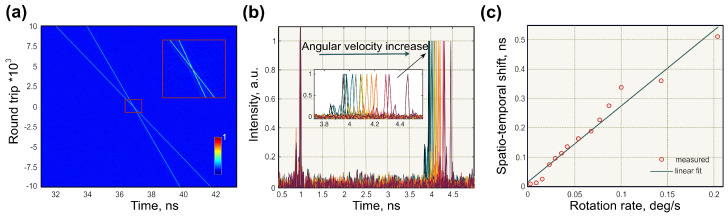
Optical Sagnac effect detection through spatiotemporal pulse evolution: (**a**) the spatiotemporal dynamics of counter-propagating pulses with different repetition frequencies; (**b**) temporal shift of combined pulses after 10${}^{4}$ round trips at different angular velocities; and (**c**) pulse temporal shift due to the optical Sagnac effect (in reference to the platform at rest) at 10 000 round trips beyond the point of pulse overlapping. Adapted from [29] *©* CC BY.

**Figure 12 sensors-21-03530-f012:**
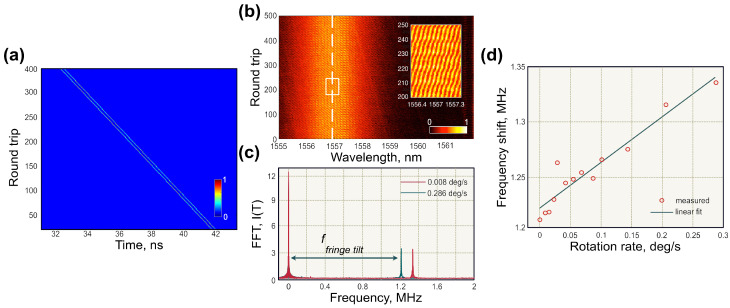
Optical Sagnac effect measurements by single-shot spectral evolution using DFT: (**a**) the spatiotemporal dynamics of counter-propagating pulses with synchronised repetition frequencies; (**b**) single-shot spectra dynamics within 500 round trips. Inset: zoomed-in central part of spectral evolution; (**c**) frequency shift for two different angular velocities; and (**d**) DFT response to different angular velocities. Adapted from [29] *©* CC BY.

**Figure 13 sensors-21-03530-f013:**
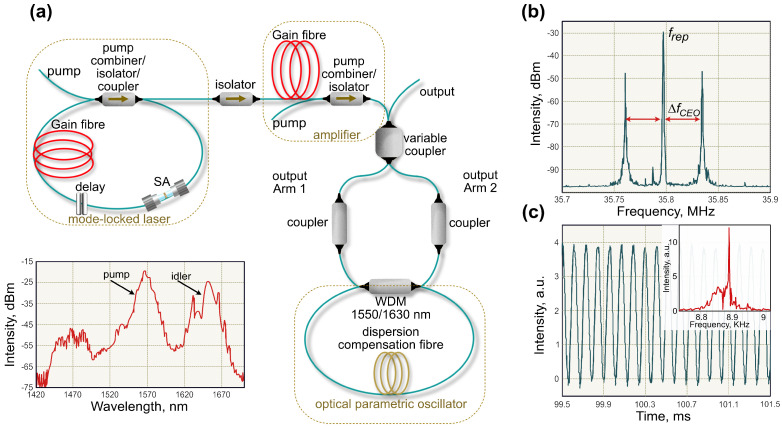
Bidirectional fibre optical parametric oscillator for rotation measurement: (**a**)schematic setup (inset: output spectrum); (**b**) RF spectrum of observed frequency beating; and (**c**) beating of the two frequency combs in time domain (inset: Fourier transform of the time domain signal, showing beat-note frequency). Adapted with permission from [152] *©* The Optical Society.

**Table 1 sensors-21-03530-t001:** Summary of different ultrafast fibre laser gyroscopes performance and data acquisition methodologies.

Cavity	Gyroscope	Measurement	Angular Velocity	Resolution	Scale Factor	Stability	Advantages	Disadvantages	Ref.
configuration	length/area	method	range			performance
NALM-based						phase noise	High stability	Sagnac effect is	
NALM-based						random walk	laser configu-	through	
with polariser	150 m/0.07 m${}^{2}$	time delay	up to 240 deg/s	0.4 deg/$\sqrt{h}$	0.47 ps/(deg/h)	0.06 deg/h	ration	roundtrips	[196]
						zero bias drift	Well-established	Low stability	
Ring cavity	4.59 m/0.79 m${}^{2}$	beat-note	0.12–90 deg/s	0.028 deg/s	7 kHz/(deg/s)	193 Hz	and simple	due to fluctuation	[123]
							method	of the CEO frequency
Ring cavity:						zero bias drift	Higher stability	Hard to achieve	
dark solitons	69.5 m/0.29 m${}^{2}$	beat-note	±16 deg/s	0.05 deg/s	3.31 kHz/(deg/s)	72 Hz	of dark solitons	generation	[199]
Ring cavity:		real-time				time jitter	100s kHz–10s MHz	Requires fast	
different rep rates	13.5 m/0.3 m${}^{2}$	spatiotemporal	0.01–0.3 deg/s	0.022 deg/s	0.885 deg/s/ns	<0.8 ps	acquisition rate	photodetectors	[29]
								and oscilloscope	
Ring cavity:		real-time					Highest resolution	Requires fast	
equal rep rates	13.5 m / 0.3 m${}^{2}$	DFT	0.01–0.3 deg/s	0.007 deg/s	0.017 deg/s/kHz	-	at high data	photodetectors
							acquisition rate	and oscilloscope	[29]
								Sensitivity to
								the CEO phase	
Optical paramet-				estimated		f${}_{CCEO}$ drift	Extremely narrow	No experimental	
ric oscillator	13.5 m/0.3 m${}^{2}$	beat-note	-	3.22 × 10${}^{-7}$ deg/s	-	10s kHz over 30 s	linewidth, possibility to scale	demonstration	[152]
				at zero f${}_{CEO}$ drift			up performance		
Hollow-core fibre						bias stability 0.15 deg/h	Low environmental	High splice losses	
optic gyroscope	5.6 m/132 cm${}^{2}$		±100 deg/s	0.05 deg/h	-	ARW 0.04 deg/$\sqrt{h}$	sensitivity	demonstration	[208]
Brillouin ring						ARW noise 0.068 deg/$\sqrt{h}$	Highest resolution	Not at the point	
laser gyroscope	1017 mm${}^{2}$	beat-note	up to 250 deg/h	5 deg/h	0.073 Hz/(deg/h)	bias stability 3.6 deg/h	Integrated design	of commercialisation	[73]

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
