# Peer review of "Rotation Active Sensors Based on Ultrafast Fibre Lasers"

_sensors, 2021, doi:10.3390/s21103530_

Round 1

Reviewer 1 Report

The paper presents a review of advances in a field of laser gyroscopes. The paper consists with historical overview of the field, detailed description of the common techniques and modern state-of-art techniques related to ultra-fast measurements exploited in fiber laser gyroscope. The paper is well-written and provides the up-to-date perspective on the development of the rotation sensors based on ultrafast fiber lasers. Here some remarks:

  • Misprint at 353 line: “.luctuations”.
  • Labeling of the 3 figures on Figure 8 is confusing. (a) seems to relate with the top figure, however, in the caption it is mentioned as (c)
  • Adding one summary table with comparison of the sensitivity, advantage and disadvantages different methods would be preferable.

To sum up, the paper should be published in “Sensors” with minor revisions of misprints.

Author Response

Q1 Misprint at 353 line: “.luctuations”.

Answer: We apologise for the typo. This has been corrected.

Q2 Labeling of the 3 figures on Figure 8 is confusing. (a) seems to relate with the top figure, however, in the caption it is mentioned as (c)

Answer: We thank the reviewer for this comment. Indeed, the confusion originated during several rounds of proofreading and optimisation of the manuscript. The mistake has been corrected.

Q3 Adding one summary table with comparison of the sensitivity, advantage and disadvantages different methods would be preferable.

Answer: We have summarised the presented results on ultrafast fibre laser gyroscopes into a table, comparing them with cutting edge results on conventional FOG and ring laser gyroscope technologies. However, due to the difference of approaches to measure the optical Sagnac effect and limited presented parameters, the table does not allow a comprehensive comparison of the gyroscopic performance. 

Reviewer 2 Report

The work: sensors-1188452 explores a technique that improves gyroscopes. Furthermore, the present manuscript has high quality, and the authors consider theory and important works; in my opinion, this is a highlight for a review. The lecture can be attractive, due to fast fiber-optic lasers are not fully explored for the gyroscope. The only complaint is that any table synthesizes the information, and any clear comparison between fast fiber laser gyroscope and passive fiber optic gyroscope is presented. Moreover, It is interesting that the authors do not use commons terms in optical fiber gyroscopes, such as temperature sensitivity, bias instability, and vibration sensitivity.  I believe the authors did a hard work and only minor comments need to be attended. As a result, the comments for the article are the follows:

This manuscript presents a compressive review of fiber optic gyroscope based on fast fiber lasers. The authors provide an extensive review considering prior works; here, they consider relevant works related to the Sagnac effect. Moreover, the organization and figures have high quality. If the authors attend some minor comments, the manuscript will be suitable for publication in SENSORS.

  •  Figure 1 shows the schematic differences between conventional Sagnac setup and laser Sagnac configurations. However, the work focuses on fiber lasers gyroscopes. It is suggested to include the fiber optic Sagnac configuration and its explanation. Here, the authors need to focus on fiber optic laser configurations. Moreover, Fig 1(b) needs to be redrawn because any input/output is clearly described. 

  • Interestingly, the authors do not consider some terms related to fiber-optic gyroscopes, such as thermal noise, bias instability, and vibration sensitivity. The authors need to describe the performance of these parameters considering fast fiber-optic laser gyroscopes. 

  •  The authors need to synthesize the information using a comparative table in terms of sensitivity and angle random walk. Here, it is necessary to highlight the fast fiber-optic gyroscopes performance; then, the comparative analysis needs to include passive gyroscopes.

  • The authors present a good discussion about fast fiber laser stability. However, they need to discuss the power and wavelength fluctuations improvement using signal processing or computational techniques. 

Author Response

Q1 Figure 1 shows the schematic differences between conventional Sagnac setup and laser Sagnac configurations. However, the work focuses on fiber lasers gyroscopes. It is suggested to include the fiber optic Sagnac configuration and its explanation. Here, the authors need to focus on fiber optic laser configurations. Moreover, Fig 1(b) needs to be redrawn because any input/output is clearly described. 

Answer: We thank the Reviewer for the comment. We have modified the figure to clarify the gyroscope output. We would like to stress that the focus of the review was to bridge advanced ultrafast fibre laser development with the field of gyroscopic applications. Therefore, the beginning of the manuscript is dedicated to the fundamentals of both areas. For this purpose, the generalised representation of fibre optic gyroscope works better than discussing fibre realisation. 

Q2 Interestingly, the authors do not consider some terms related to fiber-optic gyroscopes, such as thermal noise, bias instability, and vibration sensitivity. The authors need to describe the performance of these parameters considering fast fiber-optic laser gyroscopes. 

Answer: We appreciate the Reviewer’s comment. We have complemented the manuscript with more information on thermal noise and bias instability considering gyroscopes based on mode-locked lasers. However, thermal noise and vibration sensitivity in mode-locked fibre lasers have a similar impact as in traditional fibre-optics gyroscopes.

Q3 The authors need to synthesise the information using a comparative table in terms of sensitivity and angle random walk. Here, it is necessary to highlight the fast fiber-optic gyroscopes performance; then, the comparative analysis needs to include passive gyroscopes.

Answer: We have summarised the presented results on ultrafast fibre laser gyroscopes into a table, comparing them with cutting edge results on conventional FOG and ring laser gyroscope technologies. However, due to the difference of approaches to measure the optical Sagnac effect and limited presented parameters, the table does not allow a comprehensive comparison of the gyroscopic performance. 

Q4 The authors present a good discussion about fast fiber laser stability. However, they need to discuss the power and wavelength fluctuations improvement using signal processing or computational techniques. 

Answer: We have amended the manuscript by providing more information on the stabilisation techniques to suppress the intensity and frequency noises. 

Reviewer 3 Report

Report on “Rotation Active Sensors based on Ultrafast Fibre Lasers” by Kudelin et. al. In this contribution authors present novel application of ultrafast fibre lasers in gyroscopes sensors. They review in details properties and characterization method of ultrafast fiber lasers. Moreover the also show recent results of using ultrafast lasers in gyroscopes systems.  In my opinion this review paper is interesting, well written and deserve to by published in MDPI Sensors after minor revision.

In order to improve the quality of paper please find below potential comments and suggestion:

  1. Line 353 instead of “.luctuations” it should be rather fluctuations
  2. Line 535 “such as at 1.55 and 2 4µm” it should be rather 1.55 and 2.4 µm. Please check that
  3. Also often authors use fiber instead fibre for example lines 489,490. Please be consistent and use only one form through the paper.
  4. General comment: Can authors briefly discuss potential of laser operating in MIR region (above 2.5 µm) for gyroscope applications.

Author Response

Q1 Line 353 instead of “.luctuations” it should be rather fluctuations

Answer: We apologise for the typo. This has been corrected.

Q2 Line 535 “such as at 1.55 and 2 4µm” it should be rather 1.55 and 2.4 µm. Please check that

Answer: We would like to thank the Reviewer for spotting this typo. Indeed, 2 µm wavelength range corresponds to Tm-doped fibre laser generation wavelength range. We have made corresponding corrections.

Q3 Also often authors use fiber instead fibre for example lines 489,490. Please be consistent and use only one form through the paper.

Answer: We appreciate such high attention of the Reviewer. The use of British and American spelling has been checked thoughout the manuscript and unified.

Q4 General comment: Can authors briefly discuss potential of laser operating in MIR region (above 2.5 µm) for gyroscope applications.

Answer: Indeed, the Mid-IR generation in fibre lasers is currently a hot topic. However, we do not anticipate rapid progression in ultrafast fibre laser gyroscopes operating in Mid-IR. First of all, the Sagnac-induced phase shift is inversely proportional to the wavelength (please, refer to Eq. 3 in page 3). Therefore, for higher phase shift accumulation shorter wavelengths are more beneficial, which we already stated in the manuscript in lines 550-554.

Furthermore, the field of the Mid-IR fibre lasers is not well-matured: the systems are not all-fiberised, there are no fibre-based components (as simple as fibre couplers) to create laser configurations described in the manuscript: ring or loop-mirrors. 

Therefore, with regards to the main scope of the review manuscript on ultrafast fibre laser sensors, we cannot comment on any benefits of laser operating in the Mid-IR region. As we stated in the manuscript, we foresee the development of new measurement techniques and advanced data post-processing rather than an exploration of new wavelength bandwidth.